# Community Engagement and Psychometric Methods in Aboriginal and Torres Strait Islander Patient-Reported Outcome Measures and Surveys—A Scoping Review and Critical Analysis

**DOI:** 10.3390/ijerph191610354

**Published:** 2022-08-19

**Authors:** Courtney Ryder, Jacqueline H. Stephens, Shahid Ullah, Julieann Coombes, Nayia Cominos, Patrick Sharpe, Shane D’Angelo, Darryl Cameron, Colleen Hayes, Keziah Bennett-Brook, Tamara Mackean

**Affiliations:** 1College of Medicine and Public Health, Flinders University, Adelaide, SA 5001, Australia; 2Flinders Health and Medical Research Institute, Flinders University, Adelaide, SA 5001, Australia; 3The George Institute for Global Health Australia, University of New South Wales (UNSW), Missenden Rd., Sydney, NSW 2052, Australia; 4School of Population Health, University of New South Wales (UNSW), Sydney, NSW 2052, Australia; 5Far West Community Partnerships, Ceduna, SA 5690, Australia; 6Moorundi Aboriginal Community Controlled Health Service, Lot 1 Wharf Road, Murray Bridge, SA 5253, Australia

**Keywords:** Aboriginal and Torres Strait Islander, survey, methods

## Abstract

(1) Background: In healthcare settings, patient-reported outcome measures (PROMs) and surveys are accepted, patient-centered measures that provide qualitative information on dimensions of health and wellbeing. The level of psychometric assessment and engagement with end users for their design can vary significantly. This scoping review describes the psychometric and community engagement processes for PROMs and surveys developed for Aboriginal and Torres Strait Islander communities. (2) Methods: The PRISMA ScR guidelines for scoping reviews were followed, aimed at those PROMs and surveys that underwent psychometric assessment. The Aboriginal and Torres Strait Islander Quality Appraisal Tool and a narrative synthesis approach were used. (3) Results: Of 1080 articles, 14 were eligible for review. Most articles focused on a validity assessment of PROMs and surveys, with reliability being less common. Face validity with Aboriginal and Torres Strait Islander communities was reported in most studies, with construct validity through exploratory factor analyses. Methodological design risks were identified in the majority of studies, notably the absence of explicit Indigenous knowledges. Variability existed in the development of PROMs and surveys for Aboriginal and Torres Strait Islander communities. (4) Conclusions: Improvement in inclusion of Indigenous knowledges and research approaches is needed to ensure relevance and appropriate PROM structures. We provide suggestions for research teams to assist in future design.

## 1. Introduction

Patient-reported outcome measures (PROMs) capture perspectives of health and wellbeing and are considered an important component of measuring safety and quality in healthcare [1,2,3]. These PROMs generally take the form of a survey with set items and scales and can be administered in a variety of different ways (individually or via health professional, in person or over telephone, hard copy or online). As measures, PROMs are patient centered, as the patient, or their carer, determines their status regarding quality of life, wellbeing and effectiveness of care [2,3]. The quality and robustness of PROMs can vary dramatically [1], and their creation continues to be underpinned by Western biomedical models and constructs of health and well-being [4]. While these approaches may produce PROMs suited to the dominant population, they can produce PROMs not always applicable across other population groups, and rigorous assessment to take account of population differences is lacking [5,6]. This is also true for survey quality assurance processes, where it is not always apparent if research outcomes have been appropriately contextualized for target patients or communities. This is particularly apparent in Australia, where the Australian Commission on Safety and Quality in Health Care recommend that PROMs for widespread use should undergo rigorous psychometric assessment with end users [3].

In Australia, Aboriginal and Torres Strait Islander communities experience significant health inequities from ongoing colonization, including an inequitable burden of health research [5,7]. The Australian Institute of Aboriginal and Torres Strait Islander Studies (AIATSIS) Code of Ethics for Aboriginal and Torres Strait Islander Research and the National Health and Medical Research Council (NHMRC) ‘*Road Map 3*’ and ‘*Keeping Research on Track II*’ have created core recommendations to alleviate this burden, with a focus on leadership, self-determination, sustainability, impact and accountability in the conduct of research [8,9,10]. These documents have been critical in redirecting Aboriginal and Torres Strait Islander health research to clearly benefit community. However, in many randomized controlled trials and observational and longitudinal studies, PROMs and psychometric assessment surveys developed for the dominant Australian population continue to be used with Aboriginal and Torres Strait Islander communities [5,6,11]. If any psychometric assessment occurs (i.e., reliability or validity) during the development of the PROM or survey tool, the focus is typically on Western biomedical constructs, excluding Indigenous methodologies and knowledges [4,5,12]. 

PROM development must include such knowledge in order to be appropriately contextualized for Aboriginal and Torres Strait Islander peoples and communities. Doing so produces PROMs that are decolonized, which do not reinforce unhelpful colonial knowledge or constructs [4,5,6,13]. Rather, by focusing on strength-based approaches and important community, cultural and social factors, PROMs can then act to dismantle the deficit discourse and negative data narrative surrounding Aboriginal and Torres Strait Islander health [5,11,14]. To better understand and map the literature in this area, a scoping review was deemed the most appropriate methodological approach. [15,16,17]. The overall aim of this scoping review was to *elicit key concepts for best practice in the design and assessment of PROMs and surveys for Aboriginal and Torres Strait Islander communities*. To achieve this, we addressed three objectives: to determine the depth and breadth of existing PROMs or surveys for Aboriginal and Torres Strait Islander communities; to determine how Indigenous knowledges and community engagement contributed to the design of these measures; and to determine the level of psychometric assessment undertaken and how this was conducted.

## 2. Materials and Methods

This scoping review was underpinned by knowledge interface methodology, which is an Indigenous research methodology for new knowledge formation through mutual respect, shared benefits, human dignity and discovery between knowledge systems, research methodologies and methods [7]. The scoping review followed the PRISMA-ScR process across 27 items (Appendix A) [15] and scoping review guidelines from Peters [16,17]. Unacknowledged or defined power dynamics in this process were shifted by focusing on Indigenous knowledges and ways of working [7]. This included employing Aboriginal and Torres Strait Islander critical appraisal tools [18,19], drawing on Aboriginal and Torres Strait Islander scholars, and ensuring Aboriginal and Torres Strait Islander researchers’ autonomy over all aspects of this review. 

### 2.1. Eligibility Criteria, Study Selection and Information Sources

The lead Aboriginal researcher (CR) defined the search syntax and key terms for the scoping review, which was approved by other Aboriginal and Torres Strait Islander researchers on the team, prior to database searches (Table 1). 

Inclusion and exclusion criteria were defined using the same process, with the research team focusing on capturing studies that designed and psychometrically assessed PROMs and surveys specifically for Australian Aboriginal and Torres Strait Islander communities (Table 2). In this process, the research team decided that survey responsiveness, which is commonly used as an assessment for diagnostic tools in clinical settings, was out of scope for this review [20]. Databases in Table 1 were searched for relevant literature published between 2002 and 2022, with the search conducted on 5 April 2022. 

### 2.2. Data Items, Charting Process and Results Synthesis

Title, abstract and full text screening against key words and inclusion and exclusion criteria was undertaken by an Aboriginal (CR) and non-Aboriginal (JHS) member of the research team. Both researchers met to discuss outcomes and reach consensus on the final articles to be included for full text review.

A data charting form was created by the first author to focus on the three key questions of this review. This included using the Aboriginal and Torres Strait Islander Quality Appraisal Tool [18,19] to define key extraction items in the data charting form relating to Indigenous knowledges and community consultation. The final data charting form was approved by the research team prior to data extraction. Data items were extracted under the following three domains:*Study demographics and survey parameters*: reference, study location, focus, funding, sample size and survey parameters (target age, tool name, administration, number of items and scales).*Indigenous knowledges and community engagement*: Aboriginal leadership, methodology and methods, community consultation and governance, and strength-based analyses.*Survey design*: Validity—practicality, face, content, and construct; reliability—stability, internal consistency, equivalence (descriptions, Appendix A), sample size, administration, item number (prior to validity reduction), scale and funding.

For data extraction in the domain of Indigenous knowledges and community engagement, a three-point response criteria was used in line with the Aboriginal and Torres Strait Islander Quality Appraisal Tool [18,19]. For example, under the community consultation item, a ‘yes’ response indicated that articles cited and explained processes and approaches for consulting with Aboriginal communities for their project, a ‘partially’ response indicated that articles had cited that community consultations were undertaken but did not describe processes for this, and ‘unclear’ indicated that no information could be located on community consultation processes in the article. The term ‘unclear’ was used across all domain items with articles for this review. The research team selected the term ‘unclear’ as some researchers may have met this data item in their research but not reported on it.

Given that the scoping review aim was to examine the design and assessment of PROMs and surveys for Aboriginal and Torres Strait Islander communities, a narrative synthesis was undertaken. The three-step narrative based approach described by Petticrew and Roberts (2006) was used to critically examine and explore commonalities and differences for each study across the three defined domains [21]. Research topic yarning, an Indigenous research method for rich data exploration, was conducted amongst the research team to critically analyze and explore outcomes with an Indigenous knowledges focus for reporting and discussion [22]. This narrative synthesis included focusing on the design of PROMs and surveys with linguistic considerations. 

When surveying diverse cultural groups, linguistic, functional, and conceptual equivalences are required to validate understanding and create comparable metrics [23], with methodologies incorporating linguistically and culturally appropriate instruments [24]. This is of particular relevance in Australia where Aboriginal and Torres Strait Islander languages are numerous (160 spoken at home) [25] and structurally and grammatically diverse, and English is the language of the vast majority of academic and public institutions. The selected articles were, therefore, reviewed for explicit integration of Aboriginal and Torres Strait Islander language and translation and the identification and incorporation of any non-equivalent cultural concepts in data collection and analysis methodologies. 

## 3. Results

The electronic database search returned 1080 articles. Following duplication removal (n = 259), there were 821 articles (Appendix A). Title and abstract screening identified 17 potential articles for full text review. However, following consensus discussions, three articles were excluded for the following reasons: focus on non-Indigenous participants (n = 2) and responsiveness assessment (n = 1). The reviewers reached agreement on articles for inclusion in this review.

### 3.1. Charting—Study Demographics 

Most studies focused on survey assessment in one Australian state/territory (86%, n = 12) [26,27,28,29,30,31,32,33,34,35,36,37], however two (14%) were carried out nationally (Table 3) [5,38]. Of the articles, 36% (n = 5) focused on regional settings [30,31,32,36,37], and another two on remote locations [27,29]. Surveys were designed to target different age groups (Table 3). There were a similar number of articles focused on adults [5,30,31,33,34,37,38] and on adolescents and children [26,27,32,35,36]. Surveys and PROMs were designed to focus on a range of areas spanning racism (14%, n = 2) [37,38], out-of-pocket healthcare expenditure (n = 1) [5], health-related quality of life (n = 1) [26] and health and wellbeing (29%, n = 4) [30,31,35,36].

### 3.2. Charting—Indigenous Knowledges and Community Engagement

All articles contained Aboriginal leadership, evident through the authorship list and, in some cases, mentioned in the methods [5,26,27,28,29,30,31,32,33,34,35,36,37,38]. Community consultation through informant meetings, workshops, focus groups or pilot testing occurred with Aboriginal-community-controlled health organizations, Aboriginal Elders, senior community representatives, families, Aboriginal health practitioners/workers or researchers in 71% (n = 10) of studies. Only a small number of articles (n = 4) cited how community consultation changed either their research process or items in their survey/PROM [26,31,36,37]. A further 21% (n = 3) articles partially met this criterion; these articles cited undertaking consultation but provided no further information, or they stated that the consultation process had been published in a different but related publication [34,35,38]. Almost half (46%, n = 6) of articles referred to establishment of a community reference, steering or governance committee to inform their research project [5,26,27,28,30,38]. In the remaining articles no information was provided, so it was unclear if the research had Aboriginal and Torres Strait Islander research governance processes. 

Three articles clearly explained research methodology and methods focused on Indigenous knowledges, namely the ‘*Shared Space*’ approach [30], *kurunpa* (spirit) in psychosocial stress and depression [31] and knowledge interface methodology, which brings together psychometric assessment processes with Indigenous knowledges [5]. A further four articles partially met this criterion, drawing on key Indigenous scholars’ work for definitions or using tools developed by Aboriginal researchers [26,28,32,37]. In the remaining articles (n = 7) it was not clear if any Indigenous methodologies or methods had been central or part of the research process [27,29,33,34,35,36,38]. Half the articles (50%, n = 7) took a strength-based approach in their analyses, rather than focusing on a deficit-based discourse [5,26,30,31,32,33,38]. This included acknowledging the impacts of colonization and Western biomedical knowledge systems and focusing on the capacity, resilience and sovereign rights of Aboriginal and Torres Strait Islander communities in the design and development of surveys. Other articles (n = 6) partially met this criterion, with some strength-based elements but an overall predominant focus on quantitative psychometrics in their analyses and discussion [27,28,34,35,36,37].

### 3.3. Charting—Survey Design

Psychometric validation of surveys and PROMs was the most frequently used method of assessment (85%, n = 12) (Table 4) [5,26,28,29,30,31,32,33,35,36,37,38]. Face validity was also common and used to validate 57% (n = 8) of articles in a variety of ways: focus groups, key informants, subject expects or pilot testing [26,28,30,31,32,36,37,38]. Only half the articles undertook content validity, doing so through key informants [5,26,31,32], focus groups [37] or pilot testing [28,30]. Two articles used assessment with the content validity index [5,26], and these were the only two to undertake both validity and reliability as psychometric assessments.

Exploratory factor analysis (EFA) was the main method used to perform construct validity (64%, n = 9) [5,26,28,30,31,32,33,36,38]. The suitability of data for factor analysis was assessed with both Kaiser–Meyer–Olkin sampling adequacy and Bartlett’s test of sphericity in 44% (n = 4) of articles [5,31,32,33]. Verification of factor structure from predefined variables was conducted in 29% (n = 4) articles using confirmatory factor analysis (CFA) [29,32,35,38], with two of these articles previously employing EFA (Table 4) [32,38]. Internal consistency (reliability) was confirmed with use of Cronbach’s α assessment in almost all articles (93%, n = 13) [5,26,27,28,29,30,32,33,34,35,36,37,38].

Psychometric reliability testing was undertaken in only three articles [5,26,27]. These articles used test–retest processes, with assessment periods of 2–3 days [27], 14 days [5] and 4 weeks [26] to assess survey stability. The median sample- (participant number) to-item ratio for all 14 PROMs was 9.71, with 50% (n = 7) of articles meeting or exceeding this median [27,28,30,35,36,37,38]. Half (n = 7) of the articles had a sample-to-item ratio above the recommended 10:1 split [27,28,30,35,36,37,38]. The number of items measured in a survey or PROM varied from 6 to 168, with a median of 28 items across the 14 articles. Likert or point scales across five (50%, n = 7) [26,27,29,30,33,34,37] or four items (35%, n = 5) [28,30,32,36,38] were the most common scales used. However, one article used a variety of scale types (i.e., polytomous and ordinal) [5].

The administration of surveys or PROMs in studies was mostly reported as self, parent or carer administration (n = 7, 50%) [5,26,28,29,33,38]. Another five studies (36%) used a research assistant for administration through structured interviews, reading questions aloud or directly filling in the survey for this process [27,30,32,34,36]. Where this was performed, four of these articles cited employment of an Aboriginal research assistant for this work [27,30,32,39]. The majority (79%) of studies were supported by category 1 funding, specifically the NHMRC [5,26,27,29,31,32,34,36,37,38]. 

The linguistic analysis showed that despite language being a fundamental expression of Aboriginal and Torres Strait Islander community identity, encapsulating the unique and diverse cultural concepts of health care specific to each language group, it was rarely included as a parameter or potential source of alternative concepts in survey methodologies. Equally, there was limited exploration of the impact of designating English, a key tool of colonization used to implant Western values and eliminate Aboriginal and Torres Strait Islander culture, as the gatekeeper language into which all data were translated, collected and analyzed. Of the limited studies (n = 3) that included some linguistics, focus was on removing items that would not linguistically translate [36], item modification to reflect Aboriginal phrasing [31] or cross-cultural translation between languages to ensure item meanings remained true [30].

## 4. Discussion

The outcomes from this scoping review highlight the importance of culturally relevant psychometric assessment and community engagement when developing and testing PROMs and surveys for use in Aboriginal and Torres Strait Islander communities. Most articles reported some form of community engagement, however levels of engagement and the methods used varied significantly. Exemplary actions reported within articles (which should inform all research approaches) included Aboriginal- and Torres-Strait-Islander-researcher-led work, the establishment of Aboriginal and Torres Strait Islander research governance groups, the employment of Aboriginal and Torres Strait Islander researchers and extensive local Aboriginal and Torres Strait Islander community consultations. All these actions meet core national recommendations for Aboriginal and Torres Strait Islander research [8,9,10]. However, unfortunately there were articles where it was not clear if any consultation or engagement had occurred with Aboriginal and Torres Strait Islander communities or experts. It may be that the research teams did not realize the importance of reporting these approaches, or it could also indicate that these processes were not undertaken. Nonetheless, a lack of clarity in this area does little in shaping best practice approaches and demonstrating to the industry how to lead by example and support national recommendations for Aboriginal and Torres Strait Islander health research. It also acts to reinforce dominant colonial and biomedical knowledge systems in Aboriginal and Torres Strait Islander health research, especially for PROM design and psychometric assessment, by implying this is the only way to conduct research [40,41]. 

Engagement with Indigenous knowledges (via methodology, methods or narratives) allows research inquiry and analyses to appropriately contextualize and ensure relevance for Aboriginal and Torres Strait Islander communities. Critically, it shifts the power dynamics and provides space for engagement with constructs allowing for new knowledge formation through mutual respect, shared benefits, human dignity and discovery so that strength-based processes can ensue [4,5,41]. These approaches facilitate the centralizations of the cultural determinants of health (i.e., sovereignty, connectedness, social development) that act to elevate and enhance health and wellbeing for Aboriginal and Torres Strait Islander peoples [4,5,42,43]. This also acts to counter the maintenance and reinforcement of unhelpful or unwarranted colonial constructs (i.e., laziness, no regard for health, abusive, uneducated) [44]. Development of PROMs and surveys for Aboriginal and Torres Strait Islander communities must be done with communities. In this context, Indigenous knowledges must be embedded into the methodologies and methods, with guidance from key documents such as the AIATSIS Code of Ethics, NHMRC Road Map 3 and local context documents like the South Australian Aboriginal Health Research Accord [4,7,8,9,10,45]. Research processes and actions of this nature allow strength-based decolonizing approaches to be undertaken from the start, and ensure the research needs of Aboriginal and Torres Strait Islander communities are central [4,7].

For PROMs or surveys, design or modification, face validity with Aboriginal and Torres Strait Islander communities was commonly undertaken at the initial stages of construction or refinement. This was a significant strength in the psychometric process of these studies, particularly given that consultation with end users is the least-frequently applied validation method in psychometrics [46,47]. Despite face validity not being named as a methods process in most articles, these processes are representative of research teams ensuring PROM design was focused on Indigenous knowledges so that ethnocentric construction was avoided [4]. Face validity, however, should be strengthened through the use of other validity processes to provide comprehensive rigor and robustness [4,46,48]. One of these processes, construct validity, was less commonly undertaken and only a few studies employed measurable assessment methods such as the item-level construct validity index (I-CVI). In these studies, key informants or experts rated items against key constructs as derived from the face validity process or the relevant literature [49,50]. As with other authors in this area, we recommend any PROMs or surveys developed for Aboriginal and Torres Strait Islander communities undergo both face and content validity processes [12]. In line with national guidelines, Aboriginal and Torres Strait Islander representatives involved in these processes should be appropriately renumerated and outcomes translated back to community even in early phases [8,9,10]. Both processes should make use of a knowledge interface approach to ensure use of the construct validity index while also focusing on Indigenous constructs and definitions from the face validity process [4,5,46,49]. Most studies included in this scoping review followed recommended processes for construct validity, using EFA for newly developed surveys to determine theoretical themes and theme loading to items [51,52,53,54]. However, practicality for this assessment was reported in under half of these articles, creating ambiguity in sampling adequacy—that is, in knowing whether the PROM or survey should undergo construct validity or if it needed further modification and assessment [53]. Standard practice suggests that PROMs or surveys be assessed for their practicality and that this be reported when published rather than implied [53]. Construct validity needs to be undertaken through EFA to determine theoretical themes, which can be further confirmed through CFA [5,54].

Of note is that the validity process was the only psychometric assessment performed by most studies. Only two studies sought to undertake both validity and reliability assessment for their PROMs and surveys, with test–retest methods for stability assessment. In both studies a recommended time period of no less than two weeks, but no more than two months, was undertaken. This time frame is enough time to prevent recall bias, but allows participants to have some familiarity with the PROM or survey [55,56,57,58]. Reliability for PROMs or surveys plays a significant role in assessing if theoretical themes or item scales are stable across different time periods or in different contexts for Aboriginal and Torres Strait Islander communities [55]. This is important if tools are to be used over multiple time points to measure outcomes (i.e., longitudinal or observational studies), or if they are being used to guide clinical diagnostic reasoning processes [55]. While there are challenges in having participants undertake test–retest processes, such as dropout rates or limited recruitment, it is highly recommended that stability assessment be undertaken over a theoretically justifiable interval period with Aboriginal and Torres Strait Islander communities [55,59]. Similarly, to Newton (2015), we identify that Aboriginal and Torres Strait Islander communities and health research need strong and robust measures across all psychometric areas to improve health and wellbeing outcomes but also to minimize bias wherever possible [12,54,55].

Variation existed in the number of items, ranging from 6 to 168, included in the PROMs or surveys. Poor psychometric assessment has been associated with PROMs and surveys that take long periods of time to complete [5,56,60]. Ambiguity in item wording or scales can also impact psychometric assessment; however, in this scoping review, most included articles opted to use Likert or point-based scales for items [5,56,60]. Attention to detail is paramount in this area as end users, who in this context are Aboriginal and Torres Strait Islander patients, parents, carers or children, need to be consulted in order to provide important contextual information on the number of items to be measured, as well as the wording, construction and scale appropriateness, in addition to the administrative processes of the tool [5,12]. Aboriginal and Torres Strait Islander peoples continue to face educational inequities from colonization, such as differences in numeracy and literary skills, and commonly do not have English as a first language. These are also important considerations likely to impact on the success of a PROM or survey. For instance, a focus on mono-cultural and mono-linguistic methodologies, which was the focus in most studies, reinforces existing academic and societal structures that disempower Aboriginal and Torres Strait Islander peoples, reduce inclusivity and impoverish discourses of health and wellbeing [12]. In line with national guidelines, open dialogue and integration of Indigenous knowledges, expressed through alternative and culturally appropriate terminology, should occur [8,9,10], as this will offer opportunities in responding to the diversity of Australia’s healthcare consumers and improve outcomes in our healthcare systems. Additionally, in the selected studies, administration of PROMs or surveys by Aboriginal research assistants was only undertaken in about a third of the studies. However, administration by Aboriginal research assistants is a process that ensures that cultural and social information is part of the administration process; where practical, this approach could be implemented for all psychometric studies. Where this is not feasible, or appropriate, other processes should be included to increase diversity and participation, such as audio recordings. 

With regard to strengths and limitations, a significant strength of this scoping review was the focus on Indigenous knowledges and community engagement in PROMs and the psychometric assessment processes for PROM or survey design. The lived experience Aboriginal authors brought to the overall design and reporting of outcomes in this scoping review was also a key strength. While the focus of this article paper is on Aboriginal and Torres Strait Islander communities in Australia, outcomes are applicable to other Indigenous communities internationally. Limitations of this review include not comparing or reporting on results or outcomes of each of the PROMs or surveys developed in each of the articles. 

## 5. Conclusions

This scoping review highlights significant variability in the literature on the design, development and assessment methods employed for the development of PROMs and surveys for Aboriginal and Torres Strait Islander communities. Community engagement was evident in all studies; however, ambiguity existed in terms of the use of Indigenous knowledges, methodologies and methods. Most studies focused on the validation of their measures with Aboriginal and Torres Strait Islander communities. Reliability assessment was rarely undertaken, which raises questions about the stability of measures across time and different contexts. Throughout this review, we have provided researchers with recommendations from national best practice guidelines for the future creation and psychometric assessment of PROMs and surveys within, and for, Aboriginal and Torres Strait Islander communities. 

## Figures and Tables

**Table 1 ijerph-19-10354-t001:** Information sources and search syntax.

Databases	MEDLINE, psychINFO, Pubmed, Scopus
Syntax	1. (tool* or survey* or ‘patient reported outcome measure’ or PROM).ab, kw, ti. 2. (psychometric* or validity or validation or reliability).ab, kw, ti. 3. (Aborigin* or ‘Torres Strait Islander’ or Indigenous or ‘First Nation *’ or Koori * or Nunga or Murri or Anangu).as, kw, ti. 4. (health* or medicine or wellbeing or care or hospital).as, kw, ti. 5. 1 and 2 and 3 and 4 6. limit 5 to (English language and yr = ‘2002 − Current’)

**Table 2 ijerph-19-10354-t002:** Selection of sources of evidence.

Domain	Inclusion Criteria	Exclusion Criteria
Participants:	Australian Aboriginal and Torres Strait Islander peoples	Inclusion of any non-Indigenous Australians as participants
Study Type:	Assessment: survey, PROM, questionnaire, written tool	Diagnostic assessment, functional outcome assessment, qualitative assessment, observational based (cohort, longitudinal, prospective), retrospective study, case-control study, cross-comparison study, review, assessor application, protocol
PROM/Survey/Tool Assessment:	Validity: face, content, construct AND/OR Reliability: repeatability, internal consistency	Descriptive analyses Responsiveness: receiver operator curves
Focus:	Health: social and emotional wellbeing, addiction, resilience, identity, medicine, community health, allied health	Science, engineering, law, education, arts, botany
Language:	English only	Not in English
Dates:	Since 2002	Prior to 2002

**Table 3 ijerph-19-10354-t003:** Data items from charting the data (demographics and community). Note: data were sorted according to Indigenous knowledge and community engagement criteria.

Demographics			Indigenous Knowledges & Community Engagement			
First Author, Year	Age, Setting	Focus, Outcome Measure	Aboriginal Leadership	Methodology and Methods	Community Consultation & Governance	Strength Based Analyses
Ryder, 2021. [5]	>18, national	Health expenses, OOPHE	Yes	Yes	Yes, yes	Yes
Cairney, 2017. [30]	13–34, NT WA SA ^r^	Wellbeing, interplay survey	Yes	Yes	Yes, yes	Yes
Butten, 2021. [26]	<18, QLD	HRQoL, FirstNations-CQoL	Yes	Partially	Yes, yes	Yes
Brown, 2016. [31]	16–72, NT ^r^	Psychosocial, MHM PQ	Yes	Yes	Yes, unclear	Yes
Elvidge, 2020. [28]	Unclear, NSW	Cultural safety survey, CSS	Yes	Partially	Yes, yes	Partially
Arrow, 2021. [27]	<3, WA ^R^	Oral health, ECOHIS and CARIES-QC	Yes	Unclear	Yes, yes	Partially
Thurber, 2021. [38]	≥16, national	Racial discrimination, Mayi Kuwayu (discrimination section)	Yes	Unclear	Partially, yes	Yes
Kickett-Tucker, 2015. [32]	8–12, WA ^r^	Racial identity, IRISE_C	Yes	Partially	Yes, unclear	Yes
Gould, 2015. [33]	18–45, NSW	Smoking in pregnancy, RBC	Yes	Unclear	Yes, unclear	Yes
Paradies, 2008. [37]	>15, NT ^r^	Racism, MIRE	Yes	Partially	Yes, unclear	Partially
Thomas, 2010. [36]	16–20.5, Northern Australia ^r^	Social and emotional wellbeing, Strong Souls	Yes	Unclear	Yes, unclear	Partially
Garvey, 2015. [34]	≥18, QLD	Cancer, SCNAT-IP	Yes	Unclear	Partially, unclear	Partially
Williamson, 2014. [35]	4–17, NSW	Mental health, SDQ	Yes	Unclear	Partially, unclear	Partially
Langham, 2018. [29]	12–19, QLD ^R^	Child resilience, CYRM-28	Yes	Unclear	Unclear, unclear	Unclear

^R^: remote setting; ^r^: regional setting.

**Table 4 ijerph-19-10354-t004:** Survey design. Note: data sorted according to publication date.

Demographics	Study	Validity				Reliability			Survey				
First Author, Year	Assessment Aim	Practicality	Face	Content	Construct	Stability	Internal Consistency	Equivalence	Sample Size	Administration	Item No	Scale	Funding
Ryder, 2021. [5]	Psychometric validation and reliability	KMO & BTS	Unclear	Key informants (I-CVI)	EFA	ICC	Cronbach’s α	Kappa	40	Self-administered	168	Polytomus, ordinal, dichotomous	NHMRC
Butten, 2021. [26]	Psychometric validation and reliability	Unclear	Focus groups	Key informants (I-CVI)	EFA	ICC	Cronbach’s α	Unclear	163	Self-administered	39	Likert 5-point	NHMRC, QLD CHF
Arrow, 2021. [27]	Psychometric scale evaluation	NA	NA	NA	NA	ICC	Cronbach’s α	Unclear	338	Research-assistant-administered	13 and 14	Likert 5- and 3-point	NHMRC, COCA
Thurber, 2021. [38]	Psychometric validation	Unclear	Focus groups and field testing	Unclear	EFA, CFA	NA	Cronbach’s α	NA	7501	Self-administered	24	Likert 4-point	NHMRC
Elvidge, 2020. [28]	Psychometric validation	KMO	Subject experts	Pilot testing	EFA	NA	Cronbach’s α	NA	316	Interview- and self-administered	23	Likert 4-point	No funding
Langham, 2018. [29]	Psychometric validation	NA	NA	NA	CFA	NA	Cronbach’s α	NA	233	Self-administered	28	5-point scale	NHMRC, Lowitja
Cairney, 2017. [30]	Psychometric validation	Unclear	Subject experts	Pilot testing	EFA	NA	Cronbach’s α	NA	842	Research-assistant-administered	40	Likert 5-point	CREREP
Brown, 2016. [31]	Psychometric validation	KMO & BTS	Key informants	Key informants	EFA	NA	Unclear	NA	186	Unclear	28	Likert 4-point	AIATSIS, ARHRF NHMRC
Kickett-Tucker, 2015. [32]	Psychometric validation	KMO & BTS	Pilot testing	Key informants	EFA, CFA	NA	Cronbach’s α	NA	229	Research-assistant-administered	40	Likert 4-point	NHMRC
Garvey, 2015. [33]	Psychometric validation	KMO & BTS	Unclear	Unclear	EFA	NA	Cronbach’s α	NA	252	Self-administered	39	Likert 5-point	ARC, NHMRC
Gould, 2015. [34]	Psychometric scale validation	NA	NA	NA	NA	NA	Cronbach’s α	NA	121	Research-assistant-administered	7	Likert 5-point	NHMRC, NHF
Williamson, 2014. [35]	Psychometric validation	NA	NA	NA	CFA	NA	Cronbach’s α	NA	717	Self-administered	25	Unclear	Unclear
Thomas, 2010. [36]	Psychometric validation	Unclear	Pilot testing	Unclear	EFA	NA	Cronbach’s α	NA	361	Research-assistant-administered	34	Likert 4-point	NHMRC
Paradies, 2008. [37]	Psychometric validation	Unclear	Subject experts	Focus groups	PCA	NA	Cronbach’s α	NA	301	Interview- and self-administered	31	Likert 6-, 7- and 5-point scale	NHMRC

NA: Not applicable and was not part of the overall aim. KMO: Keiser–Meyer–Olkin measure of sampling adequacy. BTS: Bartlett’s test of sphericity. AIATSIS: Australian Institute of Aboriginal and Torres Strait Islander Studies. ARC: Australian Research Council. ARHRF: Australian Rotary Health Research Fund. COCA: Colgate Oral Care Australia. CREREP: Cooperative Research Centre for Remote Economic Participation. NHMRC: National Health and Medical Research Council. NHF: National Health Foundation. QLD CHF: QLD Children’s Hospital Foundation.

## Data Availability

Not applicable.

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
