# Peer review of "Community Engagement and Psychometric Methods in Aboriginal and Torres Strait Islander Patient-Reported Outcome Measures and Surveys—A Scoping Review and Critical Analysis"

_ijerph, 2022, doi:10.3390/ijerph191610354_

Round 1

Reviewer 1 Report

This paper is described as a scoping review "to better understand and map the literature in this area" and refers to Pham et. al. (2014) as the primary citation for what a scoping review entails. Unfortunately, Pham et. al. concluded that there is no agreed-upon definition of a scoping review, saying "because of variability in their conduct, there is a need for their methodological standardization to ensure the utility and strength of evidence."  Nonetheless, they said that "scoping reviews aim to provide a descriptive overview of the reviewed material without critically appraising individual studies or synthesizing evidence from different studies." (emphasis added). However, as indicated in the concluding lines of this article (lines 389-390) "throughout this review, we have provided researchers with recommendations..." The article indicates what is considered "exemplary" as well as noting what "must be done" or "should be strengthened" or what research "should make use of." While the article purports to "better understand and map the literature in this area" (i.e., a scoping review), critical appraisal in this article starts in the Introduction, such as "PROM development must include such knowledge." (line 66).

This paper is a well-written evaluation and critique of the development and psychometric methods for PROM used with Aboriginal and Torres Strait Islander populations, but it is not a scoping review. Mixing a descriptive review with an evaluation results in findings like “exemplary actions reported within articles” (line 265) that reflect the pre-defined standards that were used as selection criteria. The paper includes useful expert commentary and important concepts by the authors, but those characteristics are more appropriate for a critical review than a scoping review.

As a critique, the article could use terms such as “ongoing colonization” or “dominant Australian population” or “key tool of colonization.” Focusing on the 1.7% of identified articles that met all the inclusion criteria could be part of the critique. However, a review that excludes the vast majority of studies that initially qualified for inclusion has to show how the results are still representative of the literature (i.e., external validity of the review conclusions). If there was a comparison between a broad review of the relevant literature and the results obtained using the strict criteria presented in the article, that would be useful in addressing that limitation.

Given that it is an evaluation, the article should clearly describe the evaluation criteria. Is it based on the NHMRC standards that are briefly mentioned? Is it based on the opinions of the authors, which are not clearly stated? The article design, mixing review with critique, may be reflective of the shifting of "unacknowledged or defined power dynamics" (lines 86-87), but that is not well-defined. Without clearly describing the standards being used for evaluation, or depending on the authors prior work (Ryder et. al. , 2020), the article has the same qualities that earned marks of "partially" or "unclear" in the review.

This paper is a well-written evaluation and critique of the development and psychometric methods for PROM used with Aboriginal and Torres Strait Islander populations, but it is not a scoping review.

Given that it is an evaluation, the article should clearly describe the evaluation criteria. Is it based on the NHMRC standards that are briefly mentioned? Is it based on the opinions of the authors, which are not clearly stated? The article design, mixing review with critique, may be reflective of the shifting of "unacknowledged or defined power dynamics" (lines 86-87), but that is not well-defined. Without clearly describing the standards being used for evaluation, the article has the same qualities that earned marks of "partially" or "unclear" in the review. 

Reviewer 2 Report

The article points out the importance of culturally relevant psychometric assessment and community engagement when developing and testing patient reported outcome measures. Relevant but rarely found for indigenous communities/populations. This article, when published, will also be a valuable reference and recommendations for others' future research on patient reported outcomes, and for planning own research design.

It will be beneficial to make a mention to following article (in discussion or in introduction section) as additional information what was published in past:

Newton D, Day A, Gillies C, Fernandez E. A review of Evidence‐based evaluation of measures for assessing social and emotional well‐being in Indigenous Australians. Australian Psychologist. 2015 Feb 1;50(1):40-50.

This is review article, aim was to examine design and assessment of PROM and surveys for Aboriginal and Torres Strait Islander communities. This review describes the selected published articles presenting construction of PROM instruments; the psychometric validation and community engagement processes for PROM and surveys developed specifically for Aboriginal and Torres Strait Islander communities.

The article points out the importance of culturally relevant psychometric assessment and community engagement when developing and testing patient reported outcome measures for specific indigenous populations/communities. Relevant but rarely found for indigenous communities/populations. This article, when published, will also be a valuable reference and recommendations for others' future research on patient reported outcomes, and for planning own research design.

For this review article, the Eligibility Criteria, Study Selection and Information Sources are adequately described (in Table 1 & 2)

The conclusion is consistent with the evidence and arguments in the review.

Table 3. Data Items from Charting the Data (Demographics and Community)

-         instead sorting by year, recommend sorting according to Indigenous Knowledges & Community Engagement criteria, placing first articles that met all criteria and so on to the last that met least criteria.

Round 2

Reviewer 1 Report

The revisions and additional information in support of the methods and procedures were useful and the cover letter provided sufficient context for reconsideration.

Author Response

Please see correspondence to the Review Editor for tracked changes in this document. 
